# A New Approach for Quantifying Purpurogallin in Brewed Beverages Using LC-MS in Combination with Solid Phase Extraction

**DOI:** 10.3390/foods11101429

**Published:** 2022-05-16

**Authors:** Yu-Chen Liao, Taejo Kim, Juan L. Silva, Bang-Yuan Chen

**Affiliations:** 1Department of Food Science, Nutrition and Health Promotion, Mississippi State University, Mississippi State, MS 39762, USA; stevenliao0119@gmail.com (Y.-C.L.); jls46@msstate.edu (J.L.S.); 2Department of Kinesiology, Health, Food and Nutrition Sciences, University of Wisconsin-Stout, Menomonie, WI 54751, USA; kimt@uwstout.edu; 3Department of Food Science, Fu Jen Catholic University, No. 510, Zhongzheng Road, New Taipei City 242, Taiwan

**Keywords:** antioxidant, brewed beverages, purpurogallin, roasting, solid-phase extraction

## Abstract

Purpurogallin (PPG) is a phenolic compound known for its high antioxidant properties in plant-based food materials. However, there is no easy and reliable method for direct determination of PPG in brewed beverages owing to its hydrophobicity, which makes it hard to separate from the background hydrophobic components. Therefore, a method employing solid-phase extraction (SPE) and liquid chromatography-mass spectrometry (LC-MS) was developed for detection and quantification of PPG in brewed beverages, and PPG content was quantified in commercial coffee, cocoa, and tea samples. The limits of detection and quantification were 71.8 and 155.6 ng/g dry weight (dw), respectively. The recovery with SPE was 26.6%. When combined with acetonitrile extraction (ANE), the recovery was 6.8%, higher than 2.6% with water extraction (WTE). Test tube extractions were better than moka pot brewing (MPB) for PPG quantification. Total PPG content of ground coffees prepared by ANE, WTE, and MPB ranged between 635 and 770, 455 and 630, and 85 and 135 ng/g dw, respectively. PPG was detected in two English breakfast tea samples (335–360 ng/g dw) using WTE, but not in cocoa samples. ANE showed higher (*p* < 0.05) PPG levels, but WTE (*r* = 0.55, *p* < 0.01) correlated better with MPB than ANE (*r* = 0.43, *p* < 0.01). The result indicated that WTE is the best method to determine PPG in brewed beverages. This work demonstrated that PPG was significant in brewed coffee, and our pioneer study in developing the method for beverage sample preparation and LC-MS analysis has made possible industrial applications and provided new perspectives for future research.

## 1. Introduction

Coffee, one of the most popular beverages in the world, is a major contributor of antioxidant polyphenols in the diet [1]. Both green and roasted coffee beans have shown hepatoprotective [2], anti-inflammatory [3], and cancer-inhibiting [4] effects in animal studies. Human epidemiological studies also showed that moderate coffee consumption offered cardiovascular benefits [5] and led to lower risk for some chronic diseases including type 2 diabetes [6]. In the food industry, coffee extracts can be used to protect processed food against lipid oxidation [7] as well as regenerate the antioxidants bound to the insoluble components of foods [8].

Roasting is a heating process that turns green coffee beans into dark brown ones, which provides body, color, flavor, and aroma to the brewed coffee. Maillard reaction products are the prevailing antioxidants in roasted coffee since free chlorogenic acid is lost [9]. Antioxidant activity decreases with roasting level, but the formation of melanoidin during roasting could also improve coffee antioxidant capacity [10]. There is no conclusive evidence regarding the relationship between coffee roasting and its total antioxidant capacity [11,12,13,14].

Purpurogallin (PPG; 2,3,4,5-tetrahydroxybenzo[7]annulen-6-one, CAS 569-77-7) (Figure 1) is an active cytoprotector found in cabbage, glacial algae, chestnut oak bark, nutgalls, and leaves [15,16,17]. PPG has high peroxyl radical scavenging activity (the relative oxygen-radical absorbance capacity value was 6.01 ± 0.42) [18], and it is the first intermediate formed from pyrogallol through autoxidation or enzymatic oxidation [19]. In cell culture studies, PPG has inhibited nitric oxide synthesis and cancer cell growth [20], and prolonged cardiocytes survival against different oxidants [21]. In the food industry, PPG can be used as an additive to edible and non-edible oils or fats to retard oxidation [22]. In addition, developing more efficient conversion techniques for PPG synthesis using microbes, enzymes, and chemicals is a hot topic because of the favorable biochemical properties of PPG [23].

Pure PPG and PPG produced using whole-cell transformation can be analyzed and quantified using high-performance liquid chromatography (HPLC), and then further confirmed using UPLC-QE-MS [23]. The spectrophotometric measurement of absorbance at 420 nm can also be used for detection of purpurogallin [24]. However, background subtraction cannot be performed accurately for food samples and thus the method is not reliable. To our knowledge, it has not been identified or quantified in brewed beverages such as coffee before and it is very difficult to detect and quantify PPG in foods, given the interference of hydrophobic background components. Solid phase extraction (SPE) is a popular, rapid, selective, and versatile sample clean-up technique to HPLC and LC-MS analysis [25], and it does not require a trained analyst and relatively large volumes of organic solvents as traditional extraction (LLE) is employed [26]. The hydrophobic background problem may be resolved by using liquid chromatography-mass spectrometry (LC-MS) for mass-to-charge ratio (*m/z*) separation in combination with SPE C_18_ cartridges for sample preparation.

The objectives of this study were: (1) to develop an analytical method for detection and quantification of PPG in beverages; (2) to compare the performance of different sample preparation procedures; (3) to quantify PPG content in commercial coffee, cocoa, and tea samples; and (4) to determine PPG concentration in coffee samples labeled with different roasting levels.

## 2. Materials and Methods

### 2.1. Chemical Reagents and Beverage Materials

Purpurogallin (PubChem CID: 5281571) from TCI (Tokyo Chemical Industry Co., Tokyo, Japan) was used to prepare external standard. Solvents and chemicals were purchased from Sigma-Aldrich (St. Louis, MO, USA) and Fisher Scientific (Pittsburgh, PA, USA). All solvents that were used were HPLC grade. Commercial prepackaged ground coffee (total: *n* = 34; light: *n* = 2; medium: *n* = 16; medium/dark: *n* = 6; dark: *n* = 10), partially decaffeinated ground coffee (total: *n* = 3; medium: *n* = 2; medium/dark: *n* = 1), decaffeinated ground coffee (total: *n* = 7; medium: *n* = 5, medium/dark: *n* = 2), instant coffee (total: *n* = 7; light/medium: *n* = 1; medium: *n* = 3; medium/dark: *n* = 1; dark: *n* = 2), cocoa (total: *n* = 3), and tea (total: *n* = 4) samples were purchased at random from local supermarkets and prepared as described below, and all the following experiments were performed with triplicate assays.

### 2.2. Crude Coffee Extract Preparation

For acetonitrile extraction (ANE), 1 g of ground coffee, instant coffee, cocoa, or tea was extracted with 9 mL of acetonitrile (AN) in an 18 × 150 mm glass test tube at 121 °C, 15 psi for 15 min using an autoclave (model MEA 109-85, Market Forge Sterilmatic, New York, NY, USA). After adding 3 mL AN and vigorous vortexing, the autoclaved tube was heated at 100 °C for 15 min in a forced-air oven (model Isotemp 318F, Fisher Scientific, Fair Lawn, NJ, USA). One milliliter of AN was added to the tube and mixed by vortexing. The liquid was collected, and centrifuged (model 5415C, Eppendorf, Hauppauge, NY, USA) at 15,982× *g* (14,000 rpm) for 5 min. The supernatant was collected and stored at 4 °C. The AN extract was diluted 10 times (*v*/*v*) with water before loading onto the SPE cartridge.

For water extraction (WTE), 1 g of ground coffee, instant coffee, cocoa, or tea was extracted with 9 mL water in an 18 × 150 mm glass test tube at 121 °C, 15 psi for 15 min using an autoclave (model MEA 109-85, Market Forge Sterilmatic, New York, NY, USA). After vigorous vortexing, the autoclaved tube was heated at 100 °C for 15 min in a forced air oven (model Isotemp 318F, Fisher Scientific, Fair Lawn, NJ, USA). The liquid was collected, and centrifuged at 15,982× *g* (14,000 rpm) for 5 min (model 5415C, Eppendorf, Hauppauge, NY, USA). The supernatant was collected and stored at 4 °C.

For moka pot brewing (MPB), 5 g of ground coffee was brewed with 50 mL water in a 1-cup stovetop moka maker (model moka express, Bialetti, Rancho Cucamonga, CA, USA) on a stirring hot plate (model 11-500-49SH, Fisher Scientific, Pittsburgh, PA, USA) until the top of the pot was full of coffee (approximately 5 min). Coffee was collected and centrifuged at 38,759× *g* (18,000 rpm) using a Fiberlite F21-8X50Y rotor for 5 min at 25 °C (model Sorvall LYNX 4000, Thermo Scientific, Langenselbold, Germany). The supernatant was collected and stored at 4 °C.

### 2.3. Solid Phase Extraction (SPE)

A cleanup of the extract was conducted by using a Sep-Pak Classic C_18_ syringe-barrel-type cartridge (Waters, Milford, MA, USA) which was conditioned with 3 mL AN followed by 3 mL water. The extract was directly loaded onto the conditioned cartridge, and the procedure was accomplished under normal atmospheric pressure. For protocol optimization, the eluent was collected in a series of fractions using 1 mL of 0 to 100% (*v*/*v*) AN in water in 10% increments. For separation of PPG, the cartridge was washed with water (3 × 3 mL) followed by 20% (*v*/*v*) AN in water (3 × 3 mL) to remove polar compounds. The cartridge was eluted with 1 mL of 80% (*v*/*v*) AN in water to avoid elution of superhydrophobic compounds. The eluent was passed through a 0.45 µm Nylon membrane filter (model Millex-HN, Millipore, Billerica, MA, USA) before being analyzed by LC-MS.

### 2.4. LC-MS

The SPE eluents were analyzed using an Agilent 1200 series HPLC system coupled with an Agilent 6410 triple quadrupole mass spectrometer with an electrospray ionization (ESI) source (Agilent Technologies, Palo Alto, CA, USA). A 1.7 µm 150 × 2.1 mm C_18_ column (model Kinetex, Phenomenex, Torrance, CA, USA) was used for chromatographic separations at 30 °C. The injection volume was 5 μL for all standards and samples. An isocratic mobile phase of AN/H_2_O (70/30, *v*/*v*) with 0.1% (*v*/*v*) formic acid was used at a flow rate of 0.2 mL/min. The total run time was 8 min and the post-run time was 4 min. The negative ionization mode was used for the analytes and the capillary potential was set at 4 kV. Nitrogen was utilized both as nebulizer gas at a pressure of 40 psi (350 °C) and as drying gas at a flow rate of 10 L/min. Quantification of PPG for each sample was repeated three times (replications) by tandem mass spectrometry (MS2) scan at *m/z* 219.1 (extracted ion chromatogram, EIC) with a fragmentor voltage of 121.0 V within the range of *m/z* 100–500 at a scan time of 500 ms/cycle.

### 2.5. Quantification

External standards were used for PPG quantification. Calibration curves were created by peak areas of standards with known concentrations (0.2, 0.5, 1, 2, 5 µg/mL in 80% (*v*/*v*) AN). The calibration curves were linear and *R*^2^ values were greater than 0.98.

### 2.6. Recovery

Recovery tests were performed for the two test tube extractions (ANE and WTE) in combination with the SPE method and the SPE method alone. For the two test tube extractions, 1 mg of PPG was added to an 18 × 150 mm glass test tube with 10 mL of solvents (AN or water) and was prepared, using the pressurized-heat extraction and the SPE method mentioned above. For the SPE method, 1 mL of 100 µg/mL PPG solution of AN was prepared via the aforesaid SPE procedures. All eluents were assayed in triplicate, and the recovery rates were generated by comparing the mass spectral data to the calibration curves.

### 2.7. Statistical Analysis

Statistical analysis was carried out with the SAS for Windows software (version 9.3, SAS Institute, Cary, NC, USA). Data were analyzed by analysis of variance (ANOVA) using the general linear models (GLM) procedures. Separation of means was conducted using Duncan’s multiple range tests at α = 0.05. Pearson’s correlation coefficient was also employed to test the relationships among different extraction methods.

## 3. Results and Discussion

### 3.1. Method Development and Performance

The detection limit of LC-MS method was 71.8 ng/g dry weight (dw) and the quantification limit was 155.6 ng/g dw, which were calculated according to the method of Armbruster, Tillman, and Hubbs [27]. Since the collision-induced dissociation (CID) product ion formation of PPG was inconsistent, the EIC MS2 spectra at *m/z* 219.1 was used to monitor and quantify PPG instead of multiple reaction monitoring (MRM)-based analysis. The recovery using SPE cartridge was 26.6%, and was 6.8% and 2.6%, respectively, when incorporating SPE into ANE and WTE. The low recovery can result from the inability of the eluting solvent to displace the analytes from the ion-exchange sites [28]. However, using solvents with strong elution power can result in a strong background, and thus further optimization is needed.

Direct injection of crude coffee extracts onto the LC-MS system resulted in peak distortion, retention time shift, and serious carry-over problems (data not shown), which are common problems in MS chromatograms [29]. Using C_18_ SPE cartridges with general recommended procedures (cartridge washed with 3 × 3 mL water followed by elution with 1 mL AN) partially solves this problem and concentrates hydrophobic ingredients, including PPG. However, hydrophobic ingredients of a preceding sample still had carry-over effects on the following samples, which made the results inconsistent and greatly increased the post-run time.

A stepwise gradient elution was performed to optimize washing and elution conditions (Table 1). Total PPG elution rate of the first three fractions (0%, 10%, and 20% AN elution) was 1.4%, and the elution rate of the last two fractions (90% and 100% AN elution) was 0.4%. Most PPG was recovered in the fraction of 40% AN elution. This indicated that only 1.8% of PPG would be lost using the proposed SPE method as compared to 100% AN for elution.

MS2 chromatograms of PPG standards and coffee samples prepared by the general recommended SPE procedures (eluted with 100% AN) or the proposed SPE method are shown in Figure 2. The retention time (the peak maximum) of PPG was roughly 1.96 min (Figure 2A). Compared with the general SPE procedure (Figure 2B), the chromatogram of the proposed method showed less peak broadening, less decrease in peak height, and fewer peak shoulders. The proposed method improved the limit of PPG detection and quantification, and provided more consistent results.

### 3.2. PPG in Brewed Beverages

The PPG concentration of six different kinds of beverages (ground coffee, partially decaffeinated ground coffee, decaffeinated ground coffee, instant coffee, cocoa, and tea) were determined, and the effects of three different extraction methods were compared (Figure 3). Since MPB could only be used for ground coffee samples, there was no comparison with other beverages for this extraction method. In three types of ground coffee samples, ANE extracts had the highest (*p* < 0.05) PPG levels while MPB samples showed the lowest (*p* < 0.05) levels. The recovered total PPG in all ground coffee samples ranged between 635 and 770, 455 and 630, and 85 and 135 ng/g dw for ANE, WTE, and MPB, respectively (Figure 3).

All three extraction methods were positively and moderately correlated (*r* = 0.43–0.55, *p* < 0.01) with each other for all coffee samples. Boiling ground coffees in water under high pressure was shown to be the most efficient way to extract antioxidants [30]. The results demonstrated a stronger correlation (*r* = 0.55, *p* < 0.01) between WTE and MPB than between ANE and MPB (*r* = 0.43, *p* < 0.01). In addition, the PPG values were 5 times higher (*p* < 0.05) in WTE extracts than in MPB extracts (Figure 3). With minimum sacrifice in PPG recovery rate, WTE also requires fewer procedures and provides more consistent results than ANE, and thus it should be a better method to extract samples using autoclave heating.

Since PPG is only slightly water-soluble [31], solutions of PPG were usually prepared in boiling 95% ethanol. ANE had a PPG recovery rate 2.5 times greater than WTE on PPG standards, but only offered little advantage on brewed beverages. ANE showed 15.6% higher (*p* < 0.05) total PPG than WTE in ground coffee samples, approximately 30% higher (*p* < 0.05) in partially decaffeinated and decaffeinated ground coffee samples, and showed no difference (*p* > 0.05) in instant coffee samples (Figure 3). Water is considered as a green solvent that is non-toxic to both human health and the environment, while AN is a volatile and flammable solvent that can possibly lead to chronic intoxication that reduces humoral and cellular immune responses [32]. Therefore, with only little sacrifice in yields, WTE can be considered as a better method while being performed in the food industry.

The higher (*p* < 0.05) PPG recovery rate of WTE on coffee samples than on pure PPG indicated that ingredients in coffee could help extract PPG in water. The low molecular weight amphipathic solutes in brewed coffee are known to decrease the surface tension of espresso coffees [33]. WTE extracts showed that there was less (*p* < 0.05) PPG in decaffeinated coffees than in ground or instant coffees, while results from ANE showed no difference (*p* > 0.05) in PPG levels (Figure 3). This suggests that some of the amphipathic solutes might be removed during the decaffeination process.

There was no difference (*p* > 0.05) in PPG levels between ANE and WTE extracts of instant coffees because they are dried water extracts of ground coffees [34]. Although cocoa beans also need to undergo roasting during production and were reported to contain pyrogallol (1.8 µg/g fresh weight) [35], PPG was not detected in the cocoa samples (Figure 3). Fermentation and modern processing have shown to result in a decrease in antioxidants in cocoa beans [36,37], and might also lower PPG to undetectable levels.

Green tea and black tea have been shown to react well with free radicals and regenerate antioxidants in food during the digestion process [8]. However, no PPG was detected in most tea samples with the exception of two English breakfast teas (335–360 ng/g dw) prepared by WTE (Figure 3). During the manufacture of black tea, polyphenoloxidase oxidizes catechins to theaflavins, thearubigins, and theabrownins [38], and might also contribute to the conversion of PPG. The results suggest that WTE might be a better extraction method for PPG determination in tea samples, and that the distinct processing steps for English breakfast teas might be key to having detectable levels of PPG.

Fifty-six phenolic compounds in black tea were reported to be screened using a developed method coupling in vitro free radical scavenging assay with UHPLC-HRMS analysis [39]. However, the method can only be applied to hydrophilic phenolic compounds, and the relatively hydrophobic compounds were reported to be co-eluted. This implies that the method might not be applied to hydrophobic phenolic compounds such as PPG, or might face challenges in analyzing samples that contain higher hydrophobic contents such as coffees. The method developed in the present study is versatile and is more likely to be performed in the food industry without the requirement of advanced analytical instruments.

Among commercial instant coffees labeled with different roasting levels, medium/dark-roasted samples had higher (*p* < 0.05) PPG levels than both light/medium and medium-roasted samples when extracted by ANE (Figure 4A), and had the highest (*p* < 0.05) PPG concentrations when extracted by WTE (Figure 4B). Among regular ground coffees, dark-roasted samples also showed higher (*p* < 0.05) PPG than both light- and medium-roasted samples when extracted by MPB (Figure 4C). There was no difference (*p* > 0.05) in PPG concentrations among different roasting levels of partially decaffeinated and decaffeinated ground coffee samples (Figure 4A–C). Dark-roasted coffees have been reported to have the highest peroxy radical scavenging activity and thus better protect human low-density lipoprotein (LDL) from oxidation [13]. These results suggest that the increased antioxidant activity might be related to higher (*p* < 0.05) PPG contents found in darker roasted coffees.

## 4. Conclusions

The developed LC-MS method preceded by test tube water extraction and SPE was found to be reliable for PPG determination in brewed beverages. The method can be used to analyze hydrophobic phenolic compound such as PPG in brewed beverage samples that contain high hydrophobic contents, such as coffees. The method does not require more advanced analytical instruments and can potentially be performed in the food industry. To prevent the elution of unwanted background hydrophobic impurities, the recovery of the SPE procedure was relatively low (26.6%). The recovery can be improved in future studies by optimizing experimental procedures, such as adopting multiple elution steps using different elution solvents. In the present study, PPG was found to be significant in all commercial coffee samples, but it was only found to be in very small quantities in some black tea samples, and it was not found in cocoa samples. Roasting to a high degree (dark-roasted coffee) might increase PPG in coffees, whereas the decaffeination treatment appears to lower PPG content in coffees. Our pioneer study in beverage sample preparation and LC-MS analysis provides possible industrial applications and can provide insights about the method development for hydrophobic compounds in food samples.

## Figures and Tables

**Figure 1 foods-11-01429-f001:**
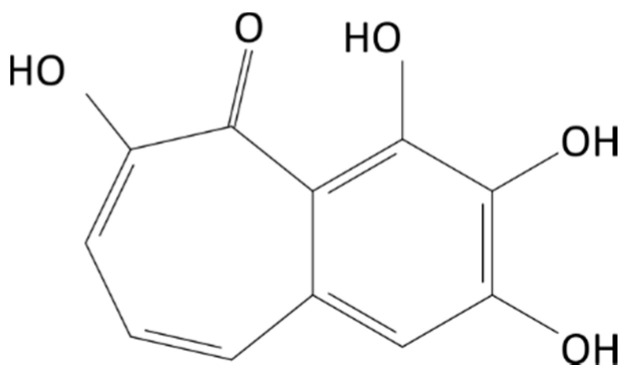
Structure of purpurogallin (PPG; 2,3,4,5-tetrahydroxybenzo[7]annulen-6-one, CAS 569-77-7).

**Figure 2 foods-11-01429-f002:**
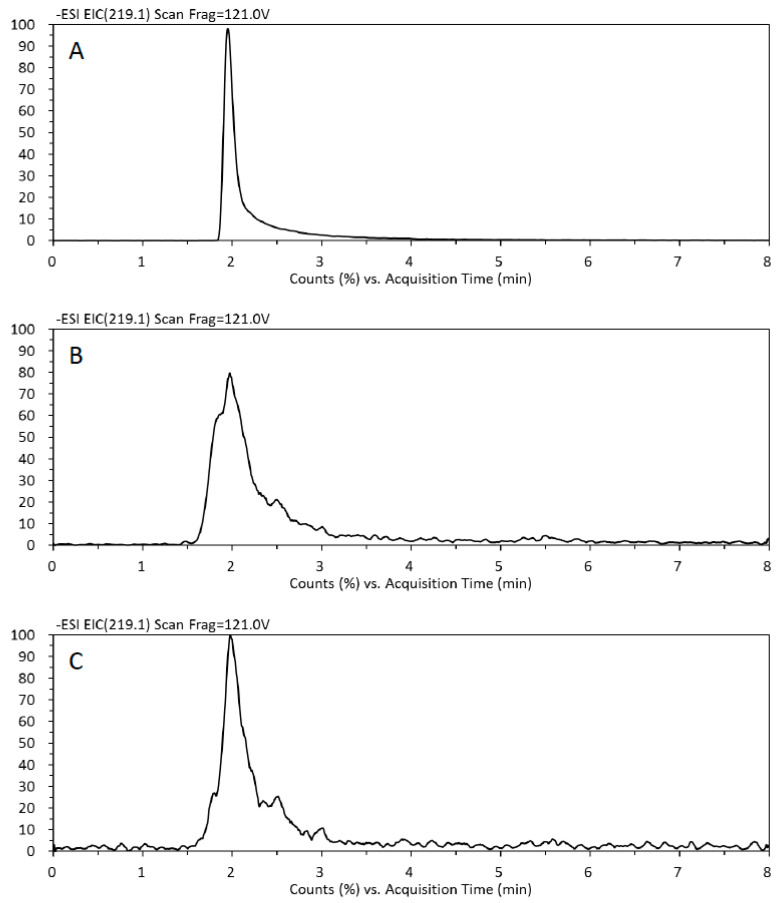
Tandem mass spectrometry (MS2) chromatograms of the purpurogallin (PPG) standard and the solid-phase extraction (SPE) eluents. (**A**) The PPG standard (1 mg/mL in acetonitrile [AN]), (**B**) the coffee eluent extracted with acetonitrile extraction (ANE) and eluted with 100% AN as the general recommended SPE procedures, and (**C**) the coffee ANE extracts prepared with the proposed SPE method.

**Figure 3 foods-11-01429-f003:**
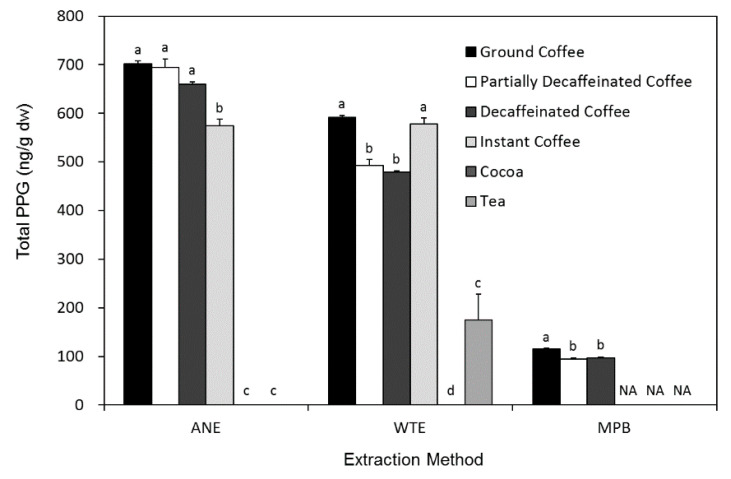
Effects of different extraction methods on total purpurogallin (PPG) analyzed by LC-MS. Data in each extraction method with different letters are different at *p* < 0.05 according to Duncan’s multiple range test, where NA denotes no data available since MPB could only be used for ground coffee samples. Data are expressed as mean ± SEM.

**Figure 4 foods-11-01429-f004:**
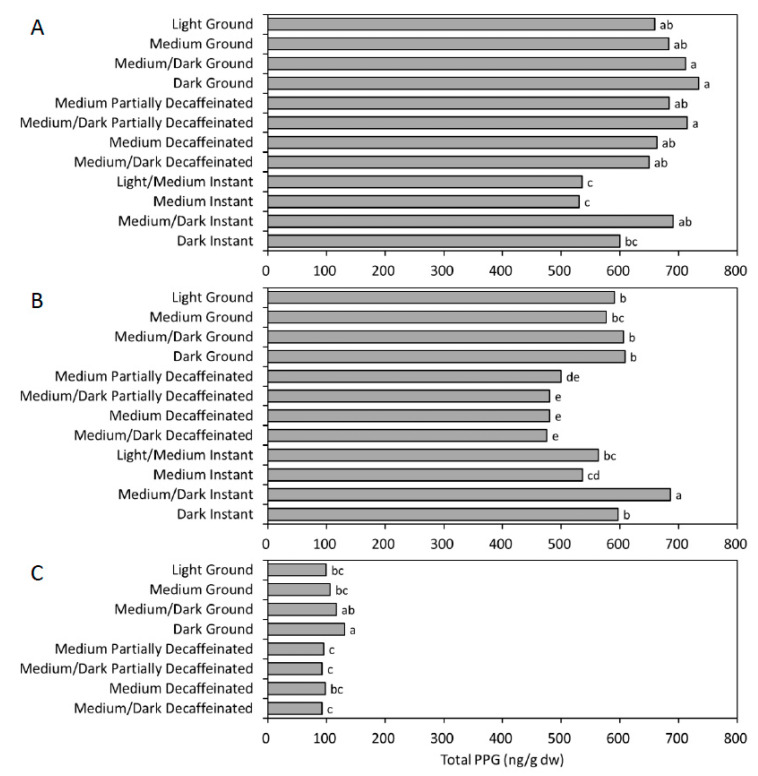
Total purpurogallin (PPG) of different roasted coffee samples extracted by: (**A**) acetonitrile extraction (ANE), (**B**) water extraction (WTE), and (**C**) moka pot brewing (MPB). Data in each chart with different letters are different at *p* < 0.05 according to Duncan’s multiple range test.

**Table 1 foods-11-01429-t001:** Stepwise gradient elution of the PPG standard (10 µg/mL in AN) from the Sep-Pak Classic C_18_ SPE cartridge.

Elution Mixture (% AN in Water, *v*/*v*, 1 mL)	Fractional Elution Rate (%)	Cumulative Elution Rate (%)
0	0.86	0.86
10	0.17	1.03
20	0.38	1.41
30	15.77	17.18
40	60.23	77.41
50	17.55	94.96
60	3.38	98.34
70	0.86	99.21
80	0.44	99.65
90	0.19	99.84
100	0.16	100.00

PPG, purpurogallin; AN, acetonitrile; SPE, solid-phase extraction.

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
