# Peer review of "A New Approach for Quantifying Purpurogallin in Brewed Beverages Using LC-MS in Combination with Solid Phase Extraction"

_foods, 2022, doi:10.3390/foods11101429_

Round 1
Reviewer 1 Report
The conducted research is scientifically significant. Research methods have been properly selected and described. The research results have been comprehensively described and presented. The work can be published in the presented form.
Detailed comments
"2.3. Solid phase extraction (SPE)"
The authors did not mention, whether the SPE procedure was carried under vacuum (using vacuum manifold) or not.
"Boiling ground coffees in water under high pressure was shown to be the most efficient way to extract antioxidants"
Line 201
Boiling ground coffee seeds.
Author Response
Detailed comments
"2.3. Solid phase extraction (SPE)"
The authors did not mention, whether the SPE procedure was carried under vacuum (using vacuum manifold) or not.
[Answer]: The SPE procedure was not accomplished under conditions of below-normal atmospheric pressure or vacuum. The Sep-Pak C18 Cartridge is a syringe-barrel-type cartridge, and the procedure was accomplished using a 5 ml disposable hand press syringe under normal atmospheric pressure. The information was added to the manuscript.
"Boiling ground coffees in water under high pressure was shown to be the most efficient way to extract antioxidants"
Line 201
Boiling ground coffee seeds.
[Answer]: We did not grind coffee beans ourselves in the present study. The coffee samples used in the experiments are commercial pre-roasted and pre-ground packaged coffee that were bought at the local supermarket. 2.1. Chemical reagents and beverage materials” in the manuscript was corrected.
Reviewer 2 Report
The authors set-up a complete new approach (extraction and analytical-HPLC-MS-methods) able to identify and quantify purpurogallin (PPG) in different coffee extracts; PPG is a phenolic compound used as a food additive with antioxidant activity. Purpurogallin topic could be interesting and literature is not so exhaustive.
See comments in detail.
Major comments
Introduction:
-Ref 22 (1994) is the only HPLC literature method? This is misleading: from a point of view this widely opens the research on this topic, from another if the research was not interested in this compound, this reduces the importance of this topic. Comment this to give significance to your work.
-Ref 22: comment the article, as this does not present an HPLC method for PPG but for 2,5-dihydroxybenzoic acid (DHBA) derivatives produced by H202.
-In addition to ref 22, the authors must add other common analytical methods to analyze purpurogallin also with other techniques (Spectroscopy?).
-Lines 51-52: if you used one ref (22) and of 1994 to assert that purpurogallin is commponly used as a food additive, this is not sufficient to explain the actual interesting on this compound. Add literature. For this, see also Minor comments (Conclusions).
-Line 61 SPE: compare the extraction procedure here proposed with other potentially present in literature and comment your choices critically. This is completely missing and must be added.
-Comment also the low recovery critically.
-Line 173 SPE: “general recommended SPE procedures” add an explanation.
Minor comments
-Line 117 mm not m.
-Conclusions: expand this part, underlying the importance of this set-up in food analysis/industry and comment the potentiality to optimize the yeld of SPE extraction procedure in future (recovery: 26.6%).
Author Response
Major comments
Introduction:
-Ref 22 (1994) is the only HPLC literature method? This is misleading: from a point of view this widely opens the research on this topic, from another if the research was not interested in this compound, this reduces the importance of this topic. Comment this to give significance to your work.
[Answer]: The new reference no. 23 and additional information are added to give more significance to the present study.
-Ref 22: comment the article, as this does not present an HPLC method for PPG but for 2,5-dihydroxybenzoic acid (DHBA) derivatives produced by H202.
[Answer]: The reference no. 22 was replaced by the new reference no. 23.
-In addition to ref 22, the authors must add other common analytical methods to analyze purpurogallin also with other techniques (Spectroscopy?).
[Answer]: The new reference no. 24 provides information about the method using spectrophotometry to measure PPG.
-Lines 51-52: if you used one ref (22) and of 1994 to assert that purpurogallin is commponly used as a food additive, this is not sufficient to explain the actual interesting on this compound. Add literature. For this, see also Minor comments (Conclusions).
[Answer]: The reference no. 22 was replaced by the new reference no. 22. The statement originally came from the patent document US 2770545 and was cited by several documents. PPG can be used as an additive to oils and fats, but it cannot be confirmed if it is commonly used in the industry.
-Line 61 SPE: compare the extraction procedure here proposed with other potentially present in literature and comment your choices critically. This is completely missing and must be added.
[Answer]: The new reference no. 25 and 26 were added to the manuscript to clarify why we chose SPE for sample clean-up in the developed method.
-Comment also the low recovery critically.
[Answer]: The new reference no. 28 was added to discuss the low recovery.
-Line 173 SPE: “general recommended SPE procedures” add an explanation.
[Answer]: The new reference no. 25 was added to support the statement.
Minor comments
-Line 117 mm not m.
[Answer]: we corrected the line 117 to “A 1.7 µm 150×2.1 mm C18 column”
-Conclusions: expand this part, underlying the importance of this set-up in food analysis/industry and comment the potentiality to optimize the yeld of SPE extraction procedure in future (recovery: 26.6%).
[Answer]: The conclusion was expanded, the potentiality of optimizing recovery in SPE procedure and the significance of the present study were added.
Reviewer 3 Report
The paper is well written in general.
Title
Suggestion to make the title catchier:
Solid phase extraction and LC-MS as an improved technique for the separation and quantification of purpurogallin in brewed beverages
Abstract
- Why the quantification of PPG is important in beverages?
- What is the best method to quantify PPG between WTE, ANE and MPB?
- Novelty of the research should be highlighted.
- An abstract should end with recommendation(s) for future studies.
- SPE should be mentioned in full in the keywords section.
Introduction
- Why the quantification of PPG is important in beverages?
- Research gap(s) should be stated. Any attempts in the previous literature to quantify PPG? What are the drawbacks of the existing methods to quantify PPG?
Materials and methods
Adequate information has been reported to allow duplication.
- How many replicates were carried out for each protocol? What is the control experiment (if any)?
Results and Discussion
- Suggestion to add the benefits of using water (green solvent) in comparison to acetonitrile.
- Comparison of the methods in this study with UHPLC–HRMS analysis can be included:
Chen, N., Han, B., Fan, X., Cai, F., Ren, F., Xu, M., ... & Yi, L. (2020). Uncovering the antioxidant characteristics of black tea by coupling in vitro free radical scavenging assay with UHPLC–HRMS analysis. Journal of Chromatography B, 1145, 122092.
Conclusion
This section has room for improvement:
- Highlight significant results
- Highlight the novelty of this research
- Include recommendations for future study
References
Please use references dating less than 6 years, from 2017 to 2022 (if possible).
Author Response
Title
Suggestion to make the title catchier:
Solid phase extraction and LC-MS as an improved technique for the separation and quantification of purpurogallin in brewed beverages
[Answer]: The topic was modified to “A new approach for quantifying purpurogallin in brewed beverages using LC-MS in combination with solid phase extraction”
Abstract
- Why the quantification of PPG is important in beverages?
[Answer]: The importance of the quantification of PPG is added to the abstract.
- What is the best method to quantify PPG between WTE, ANE and MPB?
[Answer]: WTE is the best method to determine PPG in brewed beverages. The abstract and keywords sections have been revised according to the recommendation.
- Novelty of the research should be highlighted.
[Answer]: The novelty of the research is highlighted according to the recommendation.
- An abstract should end with recommendation(s) for future studies.
[Answer]: The abstract has been revised according to the recommendation.
- SPE should be mentioned in full in the keywords section.
[Answer]: The keyword has been revised according to the recommendation.
Introduction
- Why the quantification of PPG is important in beverages?
[Answer]: The new references no. 23 and 24 provide information to elucidate the importance of the new developed method.
- Research gap(s) should be stated. Any attempts in the previous literature to quantify PPG? What are the drawbacks of the existing methods to quantify PPG?
[Answer]: The old spectrophotometry method is not reliable for food samples, and the current method required two separate steps: HPLC for quantification and LC-MS for further confirmation, which is time-consuming. Those information was added to the manuscript.
Materials and methods
Adequate information has been reported to allow duplication.
- How many replicates were carried out for each protocol? What is the control experiment (if any)?
[Answer]: All the experiments were performed with triplicate assays. The information was added to the Sec 2.1.
Results and Discussion
- Suggestion to add the benefits of using water (green solvent) in comparison to acetonitrile.
[Answer]: 1. The new reference no. 32 was added to the manuscript to provide the information.
Chen, N., Han, B., Fan, X., Cai, F., Ren, F., Xu, M., ... & Yi, L. (2020). Uncovering the antioxidant characteristics of black tea by coupling in vitro free radical scavenging assay with UHPLC–HRMS analysis. Journal of Chromatography B, 1145, 122092.
- Comparison of the methods in this study with UHPLC–HRMS analysis can be included:
[Answer]: 2. The reference was added to the manuscript as new Ref 39, and the comparison was discussed in a new paragraph.
Conclusion
This section has room for improvement:
- Highlight significant results
[Answer]: The results were highlighted according to the recommendation.
- Highlight the novelty of this research
[Answer]: The novelty of the research is highlighted according to the recommendation.
- Include recommendations for future study
[Answer]: The recommendations for future study was added to this section.
References
Please use references dating less than 6 years, from 2017 to 2022 (if possible).
[Answer]: New references were added to the manuscript according to the recommendation.
Round 2
Reviewer 2 Report
Dear authors, I saw your great work for all my previous comments and, in my opinion, now your work has been improved enough.
Thank you very much, best regards.